**SPECIAL ISSUE**
**CILIA AND FLAGELLA: FROM BASIC BIOLOGY TO DISEASE**

# *CEP290* deficiency disrupts ciliary axonemal architecture in human iPSC-derived brain organoids

Melanie Eschment[1,2,3], Olivier Mercey[4], Ellen M. Aarts[5], Ludovico Perego[1], Joana Figueiro-Silva[1], Michelle Mennel[3], Affef Abidi[6], Melanie Generali[6], Anita Rauch[1,2], Paul Guichard[4], Virginie Hamel[4] and Ruxandra Bachmann-Gagescu[1,2,3,7,*]

## ABSTRACT

Primary cilia are ubiquitous sensory organelles mediating various signaling modalities essential for development and cell homeostasis. Their dysfunction leads to ciliopathies, human disorders often affecting the central nervous system. *CEP290* is a major ciliopathy-associated gene that encodes a centrosomal and ciliary transition zone protein. CEP290 has been implicated in different cellular functions, including cell cycle control, ciliogenesis and control of ciliary membrane protein content. To investigate CEP290 dysfunction in human neurons, we generated human induced pluripotent stem cell (iPSC)-derived brain organoids harboring *CEP290* mutations. We found that CEP290 deficiency does not affect cell cycle progression or organoid formation, despite a tendency for less mature neuronal populations and formation of choroid plexus in mutant organoids. Expansion microscopy revealed morphologically abnormal ventricular cilia in the *CEP290* mutant organoid cells with bulging ciliary membranes around splayed distal axonemal microtubules. Such ciliary abnormalities might represent a tissue-specific consequence revealed by studying a human neuronal organoid model.

KEY WORDS: CEP290, Primary cilium, Ciliopathies, Brain organoids, Neurodevelopment

## INTRODUCTION

Primary cilia are nonmotile, sensory organelles that extend from the cell surface of most vertebrate cells into the extracellular space (Mill et al., 2023). During embryogenesis and brain development, primary cilia play pivotal roles coordinating signaling cascades such as sonic hedgehog and Wnt signaling (Goetz and Anderson, 2010; Wheway et al., 2018). These signals are integrated to give the required outputs in terms of gene transcription, cell proliferation, differentiation, migration, polarization or other cellular behaviors (Mill et al., 2023; Guo et al., 2015; Higginbotham et al., 2012; Bashford and Subramanian, 2022).

Cilia have a highly conserved microtubule (MT)-based structure called the axoneme, which consists of nine MT doublets extending from the base of the cilium, anchored within the cell by a specialized centriole called the basal body (BB). Located just above the BB, the transition zone (TZ) acts as a gatekeeper controlling the distinct protein composition within cilia (Reiter et al., 2012; Garcia-Gonzalo and Reiter, 2017). Proteins are actively transported in and out of cilia by the intraflagellar transport machinery (IFT), a bidirectional transport system (Lacey and Pigino, 2025). How the axoneme and ciliary tip structure are mechanically sustained is incompletely understood in primary cilia. In *Chlamydomonas reinhardtii*, *Paramecium tetraurelia* and murine photoreceptors, MTs from the basal bodies or the connecting cilium have been shown to be stabilized by an inner scaffold (Le Guennec et al., 2020; Mercey et al., 2022). In addition, MT-associated proteins have been reported to be involved in stability and ciliary tip balancing (Saunders et al., 2025; Latour et al., 2020; Conkar and Firat-Karalar, 2021).

Ciliary dysfunction leads to a group of human Mendelian disorders called ciliopathies, which present with a diverse range of phenotypes (Badano et al., 2006; Reiter and Leroux, 2017). These disorders are often associated with central nervous system (CNS) anomalies, particularly exemplified by the neurodevelopmental disorder Joubert syndrome (JBTS) which presents with a pathognomonic cerebellar and brainstem malformation on brain MRI, called the 'molar tooth sign' (Poretti et al., 2014; Bachmann-Gagescu et al., 2020). Additional CNS morphological anomalies and the presence of non-structural CNS defects, such as seizures or intellectual disability, also suggest a role for cilia in cortical neuronal function (Bachmann-Gagescu et al., 2012, 2015, 2020).

Joubert syndrome can be caused by mutations in one of >40 genes, including *CEP290*, which is one of the most intriguing ciliopathy genes. Mutations in this gene cause pleiotropic phenotypes and even distinct ciliopathies, ranging from non-syndromic retinal degeneration in Leber congenital amaurosis (LCA) to syndromic disorders including Joubert syndrome (JBTS), Bardet–Biedl syndrome (BBS) or Meckel syndrome (MKS) (Bachmann-Gagescu et al., 2015; Coppieters et al., 2010; Brancati et al., 2007; Baala et al., 2007; Wang et al., 2023). Given the importance of this gene for the spectrum of ciliopathies, a broad variety of functions has been assigned to CEP290, which makes it challenging to identify its predominant role in the CNS. CEP290 is a large protein with multiple suggested domains that have been shown to interact with various other proteins. The protein has been shown to

[1]Institute of Medical Genetics, University of Zurich, Wagistrasse 12, 8952 Schlieren, Switzerland. [2]Clinical Research Priority Program of the University of Zurich Praeclare, Wagistrasse 12, 8952 Schlieren, Switzerland. [3]Department of Molecular Life Sciences, University of Zurich, Winterthurerstrasse 190, 8057 Zurich, Switzerland. [4]University of Geneva, Faculty of Sciences, Department of Molecular and Cellular Biology, 30, quai Ernest-Ansermet, 1211 Geneva 4, Switzerland. [5]Institute of Molecular Systems Biology, ETH Zurich, Otto-Stern-Weg 3, 8093 Zürich, Switzerland. [6]iPSCore Facility, Institute for Regenerative Medicine, University of Zurich, Wagistrasse 12, 8952 Schlieren, Switzerland. [7]University Research Priority Program AdaBD, University of Zurich, Winterthurerstrasse 190, 8057 Zurich, Switzerland.

*Author for correspondence (ruxandra.bachmann@mls.uzh.ch)

M.E., 0000-0002-6881-6570; O.M., 0000-0001-7931-1642; E.M.A., 0000-0001-8714-061X; L.P., 0009-0005-6833-8168; J.F.-S., 0000-0003-3590-1784; A.A., 0000-0002-3383-9473; M.G., 0000-0001-8396-3256; P.G., 0000-0002-0363-1049; V.H., 0000-0001-5092-2343; R.B.-G., 0000-0002-3571-5271

localize to the Y-links of the TZ in *Chlamydomonas* and in retinal photoreceptors, where it was suggested to be the major component of these links, implying important roles in gating ciliary proteins (Craige et al., 2010; Mercey et al., 2024). Besides the TZ, CEP290 localizes to centriolar satellites and the BB, interacting with ciliogenesis proteins CP110 (also known as CCP110) and PCM1 (Kim et al., 2008; Tsang et al., 2008). It has also been localized in the nucleus and its loss has been proposed to affect cell cycle and to cause DNA damage (Slaats et al., 2015). Our current knowledge of CEP290 is predominantly based on studies in *Chlamydomonas,* specialized cell types, such as photoreceptors or kidney tubule cells in mouse, cell lines (e.g. hTERT-RPE1, IMCD3) and human fibroblasts or urine epithelial cells (hURECS) derived from individuals with ciliopathies (Slaats et al., 2015; Potter et al., 2021; Srivastava et al., 2017; Shimada et al., 2017; McEwen et al., 2007; May-Simera et al., 2018; Chang et al., 2006). *CEP290* mutations have been shown to variably affect ciliary length, ciliation rates and/or localization of ARL13B or ACIII (ADCY3) in various cell types (Kim et al., 2008; Slaats et al., 2015; Srivastava et al., 2017; Shimada et al., 2017; Hynes et al., 2014; Ramsbottom et al., 2018; Molinari et al., 2019). The results from these studies indicate either that CEP290 is involved in multiple key cellular processes in a given cell, or that its role varies according to cell type. Despite this abundance of data, the role of CEP290 in human neuronal cells and the pathomechanism leading from *CEP290* mutations to the observed human CNS phenotypes remain elusive. In this work, we have taken advantage of human induced pluripotent stem cell (hiPSC)-derived neuronal three-dimensional (3D) models to investigate the consequences of *CEP290* mutations in human neuronal tissues.

## RESULTS AND DISCUSSION
### Generation of a *CEP290* mutant hiPSC-derived brain organoid model
To investigate the consequences of CEP290 deficiency in human cells, we used isogenic CRISPR-gene edited hiPSC lines (Figueiro-Silva et al., 2025). We generated two mutant clones through CRISPR editing of the subcloned HMGU1 line (ISFi001-A) (Haenseler et al., 2025). The parental line was used as control (wild type; WT) along with a 'CRISPR-control' line from the same clone that had undergone CRISPR editing but did not harbor mutations at the *CEP290* target site. We targeted exon 39 of *CEP290* because many predicted truncating pathogenic variants in individuals with JBTS are found in this region of the gene (Bachmann-Gagescu et al., 2015) (Fig. S1). *CEP290* mutant 1 (ISFi001-A-1) harbors a homozygous 5-basepair deletion in exon 39 leading to a frameshift with a predicted premature stop codon 10 amino acids downstream (p.L1749Sfs*10), whereas *CEP290* mutant 2 (ISFi001-A-2) harbors homozygous five base pair missense substitutions at the beginning of exon 39 plus a homozygous 11 base pair deletion at the end of exon 39, leading to a frameshift predicted to form a premature stop codon (p.L1787Hfs*12) (Figueiro-Silva et al., 2025). Although both mutant clones are predicted loss-of-function alleles, we found residual *CEP290* mRNA lacking exon 39 due to nonsense-induced exon skipping in both lines, which appears to lead to minimal residual CEP290 protein based on western blot (Figueiro-Silva et al., 2025). This phenomenon of nonsense-induced exon skipping has been previously described in individuals with the neurodevelopmental ciliopathy JBTS (Taylor et al., 2022 preprint; Drivas and Bennett, 2014; Drivas et al., 2015), such that the lines studied here recapitulate the situation found in human disease.

Given the previously described role of *CEP290* in cell cycle control in primary mouse collecting duct cells (Slaats et al., 2015), we first evaluated whether loss of CEP290 in hiPSCs affected their cell cycle.

Flow cytometry analysis revealed no significant differences between control and *CEP290* mutant hiPSCs (Fig. S2A). We confirmed this with immunofluorescence to assess cell cycle stages using EdU and antibodies against phosphohistone 3 (PH3) which specifically marks the M phase. The percentage of EdU$^+$ nuclei confirmed flow cytometry analysis results, with 50–60% of positive cells in the S phase in all lines (Fig. S2B,C). The percentage of cells in M phase was also no different in mutants versus controls with ∼3% of PH3$^+$ cells (Fig. S2C). These results indicate that CEP290 deficiency does not affect the duration of the different stages of the cell cycle in hiPSCs.

In the developing human brain, *CEP290* is widely expressed, including in the cortical neuroepithelium, neural retina, choroid plexus and rhombic lip of the developing cerebellum (Cheng et al., 2012). This expression pattern suggests an essential role for CEP290 in brain development, consistent with the observed brain phenotypes in JBTS. To investigate the consequences of CEP290 loss in human neuronal development, we differentiated the isogenic control and *CEP290* mutant lines into brain organoids using a modified undirected brain organoid protocol (Fig. S3) (Lancaster and Knoblich, 2014). We chose not to apply more directed differentiation protocols given that these often use small molecules that directly interfere with various developmental signaling pathways such as Hedgehog (using SAG) or Wnt (using CHIR), which are known to be regulated by primary cilia (Abdelhamed et al., 2013; Wheway et al., 2013). Following neural induction, all cell lines exhibited equally homogeneous neuroepithelium formation, as evident by the presence of a thin light optical translucent ring at day 8 of culture (Fig. 1A). Brain organoids derived from all four lines expressed characteristic markers of neural differentiation, such as the neural stem cell filament marker nestin and neural progenitor marker PAX6 at day 15, as well as the mature neuronal marker MAP2 and post-mitotic neuronal marker TBR1 at day 45 (Fig. 1B,C; Figs S4, S5). Glial cells and GABAergic neurons were also present at day 80 (Fig. S6). At this stage, brain organoids from all four lines generally displayed well-organized layers in 'cortical units' surrounding a lumen, as expected with this protocol (Fig. 1D; Fig. S7A). In one of the mutant lines (mutant 1), we observed repeatedly a tendency for less-well organized or disturbed layering of such cortical units. When quantifying the thickness of the observed layers, we found a mild slightly statistically significant decrease of layer 1 (mostly TBR1+ cells) in both mutants (Fig. S7B,C). Overall, this phenotype was, however, variable between differentiation runs and not clear enough above the overall variability with this protocol to draw strong conclusions. In line with cell cycle analysis at the iPSC level and with the appearance of the expected neuronal markers over time, quantitative analysis of organoid diameter over time showed normal growth of mutant organoids on average compared to control (Fig. S8, Table S1). Taken together, these data show that hiPSCs can differentiate into brain organoids despite CEP290 deficiency with some minimal and variable differences in the organization of the primitive cortical plates.

### *CEP290* mutant organoids show increased choroid plexus formation and tendentially less mature neuronal populations
To quantitatively investigate the impact of CEP290 loss on brain organoid differentiation, we conducted single-cell RNA-sequencing (scRNA-seq) on samples of three pooled organoids for each mutant line at day 79 and two independent WT controls (72 days and 79 days of differentiation). UMAP analysis showed largely overlapping distribution of cells irrespective of genotype (Fig. 2A), with some subtle differences in proportions. Among the eight identified cell clusters (Fig. 2B), the most abundant cell type was the excitatory neurons in controls (Fig. 2C). In mutants, less differentiated neuronal

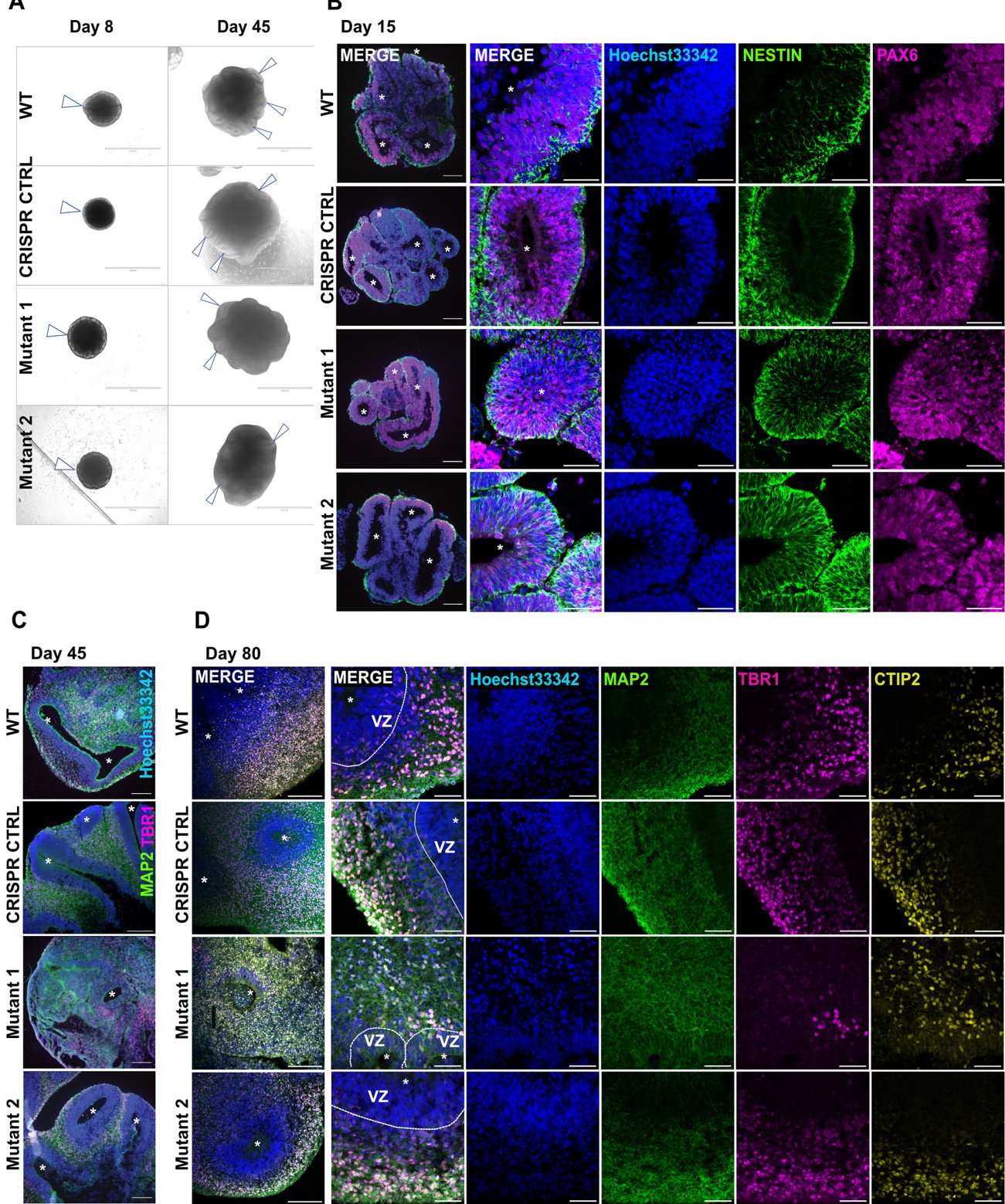

**Fig. 1.** See next page for legend.

types (radial glia and intermediate progenitors) were more abundant (Fig. 2C). In addition, we observed a cluster of transthyretin (TTR)-expressing cells predominantly present in the mutants (Fig. 2B–D). TTR is a marker for choroid plexus cells, and indeed, this finding was confirmed by the observation of TTR-positive cells organized in convoluted epithelial structures predominantly present

in mutant organoids, reminiscent of choroid plexus (TTR+ staining: 1/39 control organoids versus 11/43 *CEP290* mutant organoids) (Fig. 2D,E). Although it is surprising that the control organoids generated virtually no choroid plexus in our experiments, despite the original description of this protocol where WT organoids displayed choroid plexus-like structures over 50% of the time, this could be

**Fig. 1. *CEP290* mutant brain organoids show normal growth and develop key markers of neural differentiation.** (A) Brightfield images of control and *CEP290* mutant organoids at days 8 and 45 of differentiation. All hiPSC lines build compact cell aggregates with neural induction (arrowheads) at day 8 and exhibit visible lumen of cortical units at day 45 (arrowheads marking exemplary lumen). Scale bars: 1000 µm (day 8) and 2000 µm (day 45). (B) Overview widefield images (left panel) and confocal images (right panel) of cryosectioned control and *CEP290* mutant organoids at day 15. The separate channels for the overview widefield images are shown in Fig. S4. Immunohistochemistry of the neural progenitor marker nestin (green) and neural progenitor marker PAX6 (purple). *CEP290* mutant and control cells build comparable organoid morphologies at this stage. Scale bars: 200 µm (widefield); 50 µm (confocal). Ventricular lumen are marked by asterisks. (C) Overview widefield images of cryosectioned control and *CEP290* mutant organoids at day 45. The separate channels for these images are shown in Fig. S5. Immunohistochemistry showing the mature neuronal marker MAP2a and MAP2b (MAP2, green) and cortical layer marker TBR1 (purple). *CEP290* mutants and controls express mature neuronal- and cortical layer markers, which are located away from the ventricular lumen (asterisks). *CEP90* mutant 1 shows more disorganized and villous-like structures compared to controls and to *CEP290* mutant 2. Nuclear counterstaining with Hoechst 33342 (blue). Scale bars: 200 µm. (D) Overview widefield images (left panels) and confocal images (right panels) of cryosectioned control and *CEP290* mutant organoids at day 80. Immunohistochemistry of the mature neuronal marker MAP2a and MAP2b (MAP2, green), post-mitotic neuronal marker TBR1 (purple) and layer V/VI marker CTIP2 (yellow). *CEP290* mutant and control cells exhibit positive staining for all markers and clear structural organization in ventricular zones (VZ, marked by dashed line) and neuronal cortical layering outside of the VZ; ventricular lumen are marked by asterisks. Scale bars: 200 µm (widefield); 50 µm (confocal). All images representative of at least 3–5 organoids from at least three independent differentiations each.

explained by baseline differences in Wnt signaling in the WT (HMGU1 clone 5) iPSC line compared to the lines used in that work. Indeed, it has been shown that addition of CHIR, a Wnt agonist, leads to generation of organoids rich in choroid plexus (Pellegrini et al., 2020). The increased tendency for developing choroid plexus structures in the two isogenic lines with *CEP290* mutations could thus hint at a subtle consequence of *CEP290* deficiency on Wnt signaling. However, our RNA sequencing data did not provide strong evidence for significant dysregulation of this pathway; although a global assessment of the activity of the pathway using the PROGENy tool (Schubert et al., 2018) indicated increased Wnt pathway activity in mutant cells from cluster 7 compared to mutant cells from other clusters, individual Wnt pathway components or downstream targets were not significantly altered when comparing mutant with control cells in cluster 7 (Table S2). The conclusions that can be drawn from this analysis are however limited by the small number of control cells in this cluster. Further analysis of the scRNA-seq data using unbiased gene set enrichment analyses or targeted exploration of other signaling pathways including hedgehog signaling and analysis of expression of ciliary genes, including known CEP290 interactors, did also not reveal clear alterations to explain the observed phenotype (Tables S3–S6), and further work will be required to explore the cause for the increased choroid plexus occurrence in *CEP290* mutant organoids.

Taken together, these results show that cellular composition of brain organoids is minimally affected by CEP290 deficiency, with a trend towards less differentiated neuronal cells (radial glia and intermediate progenitors) and appearance of TTR-positive choroid plexus-like structures.

## Ventricular lumen cilia in *CEP290* mutant brain organoids are morphologically abnormal

CEP290 dysfunction has been reported to affect primary cilia in various cellular models in a cell type- and ciliopathy-dependent

manner (Slaats et al., 2015; Srivastava et al., 2017; Shimada et al., 2017; Itoh et al., 2018). However, its impact in human neuronal cells has not been described. Here, we focused on the most abundant cilia in the brain organoids located inside the ventricular lumen of the units using immunofluorescence with anti-ARL13B and anti-CEP164 antibodies (Fig. S9A). No significant differences in ciliary length were observed between controls and *CEP290* mutants at day 15 and day 30 of differentiation, and only a slight but statistically significant increase in ciliary length at day 45 existed in *CEP290* mutant 1 compared to control (Fig. S10).

Strikingly, a subset of cilia in both mutants exhibited prominent morphological abnormalities, deviating from the typical 'rod' shape of cilia. These abnormalities included bulging of the distal cilium and increased tortuosity, resulting in a 'racket-like' expansion of the tip region (Fig. 3A; Fig. S9B). These observations were confirmed by scanning electron microscopy, showing identical morphologies as seen with light microscopy (Fig. 3B). These abnormal ciliary shapes concerned only a subset of cilia analyzed by immunofluorescence but were observed in all (≥8 independent) differentiation runs each time in both mutants and not in controls. In contrast, cilia on hiPSCs displayed normal morphology (Fig. S11).

To further confirm the presence and disease relevance of these morphological ciliary abnormalities, we obtained amniocytes from a fetus carrying biallelic *CEP290* truncating variants [homozygous c.5493del, p.(Ala1832Profs*19)], reprogrammed these into hiPSCs and applied the same undirected brain organoid differentiation protocol after quality controls (Figs S12 and S13). Similar to the engineered isogenic *CEP290* mutant lines, the brain organoids generated from these hiPSCs cells grew normally, formed well-organized cortical units with established layering and similar neuronal cell types as described above (Fig. S13A,B). Importantly, identical morphological ciliary abnormalities as in our isogenic *CEP290* mutant organoids were observed for the cilia in the ventricular lumen for these organoids derived from an individual with a homozygous predicted truncating variant (Fig. S13C–E).

Ciliary tip bulging has been previously associated with deficient retrograde IFT leading to accumulation of IFT-B complex proteins at the tip (Siller et al., 2015). We were therefore interested in investigating whether IFT is affected in *CEP290* mutant organoids. We applied ultrastructure expansion microscopy (U-ExM) (Gambarotto et al., 2021) and observed that IFT88 was distributed along the cilium with accumulations at the ciliary tip and base in both controls, *CEP290* mutant organoids and the organoids derived from the fetus harboring the homozygous *CEP290* truncating variant (Fig. 3C; Figs S13D, S14). This indicates that deficient retrograde IFT is unlikely the primary cause of the observed bulging of the ciliary tip in *CEP290* mutant cilia.

To further investigate these ciliary morphological abnormalities, and to determine whether the defect concerned only the ciliary membrane or affected the underlying MT structure, we performed immunofluorescence with anti-ARL13B antibodies to mark the ciliary membrane and anti-α/β-tubulin antibodies to mark the axonemal MTs after expansion microscopy. This revealed that the axonemal integrity was affected with loss of MT coherence at the distal part, leading to splaying of the MT doublets or formation of abnormal kinks, accompanied by expansion of the ciliary membrane, in approximately half of the analyzed cilia (Fig. 3D,E; Figs S13, S15, Movie 1). In comparison, WT cilia had straightly aligned MT from the base to the tip, with the ciliary membrane tightly enclosing the ciliary axoneme (Fig. 3D). Together, these findings show that CEP290 is required to maintain axonemal MT cohesion.

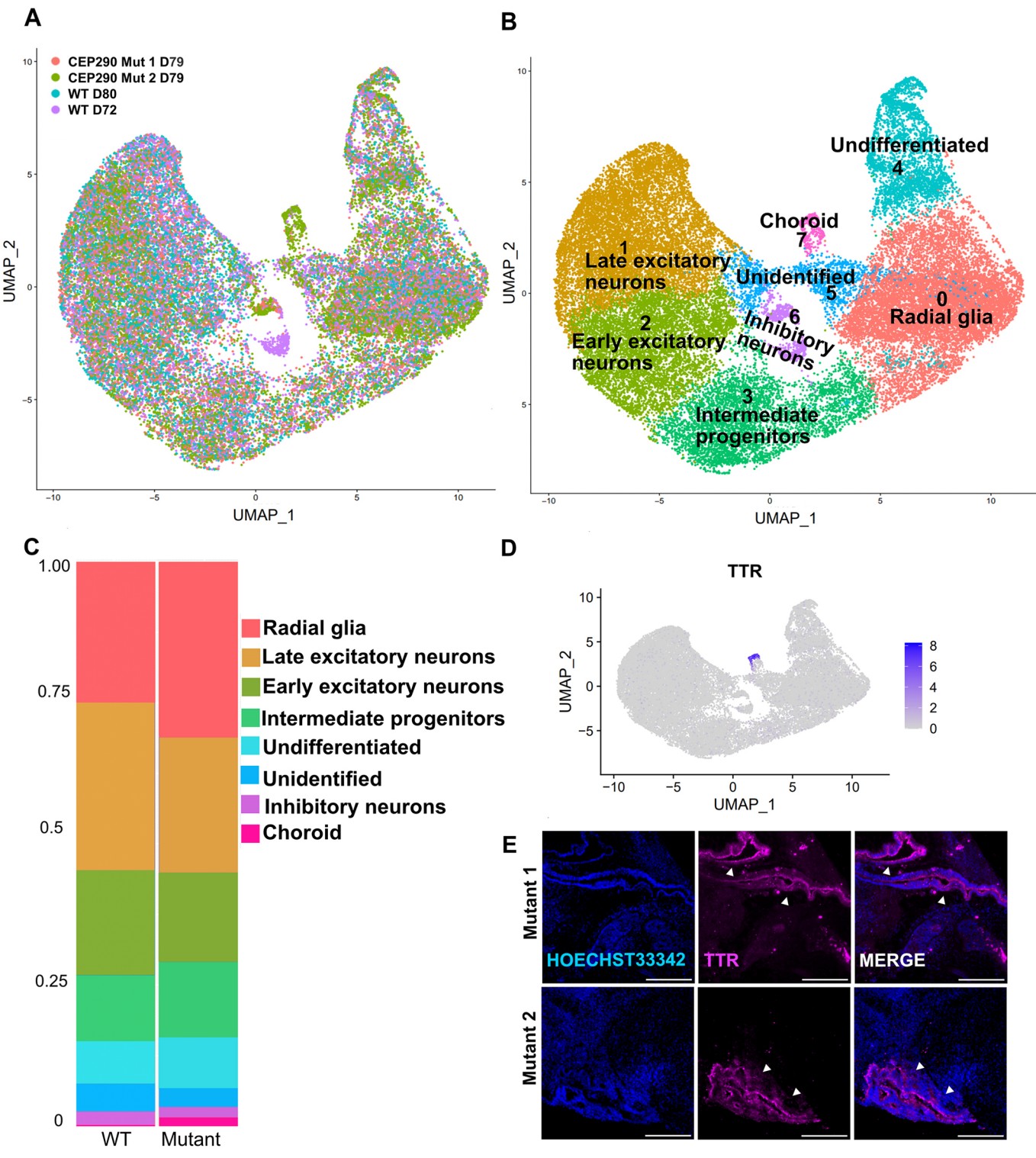

**Fig. 2. Trend towards less differentiated neurons and overrepresentation of a TTR-expressing subcluster in mutant organoids.** (A) UMAP visualization of scRNA sequencing results for two control (WT day 79 in blue and WT day 72 in purple) and two mutant lines (*CEP290* mutant 1 day 79 in red, *CEP290* mutant 2 day 79 in green). (B) UMAP plot of brain organoids from *CEP290* mutant and WT control conditions annotated by cell type identifying eight clusters. (C) Cluster distribution across mutant and control organoids shows comparable cell fractions despite mutant organoids displaying a trend towards less differentiated neuronal populations (radial glia and intermediate progenitors) and an increased subcluster of choroid plexus cells. (D) UMAP colored by expression of choroid plexus marker (TTR). A comparison with panel A indicates that these TTR-expressing cells are predominantly mutant. (E) Immunostaining of cryosections for choroid plexus at day 79. Nuclear counterstain Hoechst 33342 (blue), Choroid plexus marker TTR (purple). *CEP290* mutants show positive TTR staining, indicating presence of structures recapitulating choroid plexus (arrowheads). Scale bars: 100 µm.

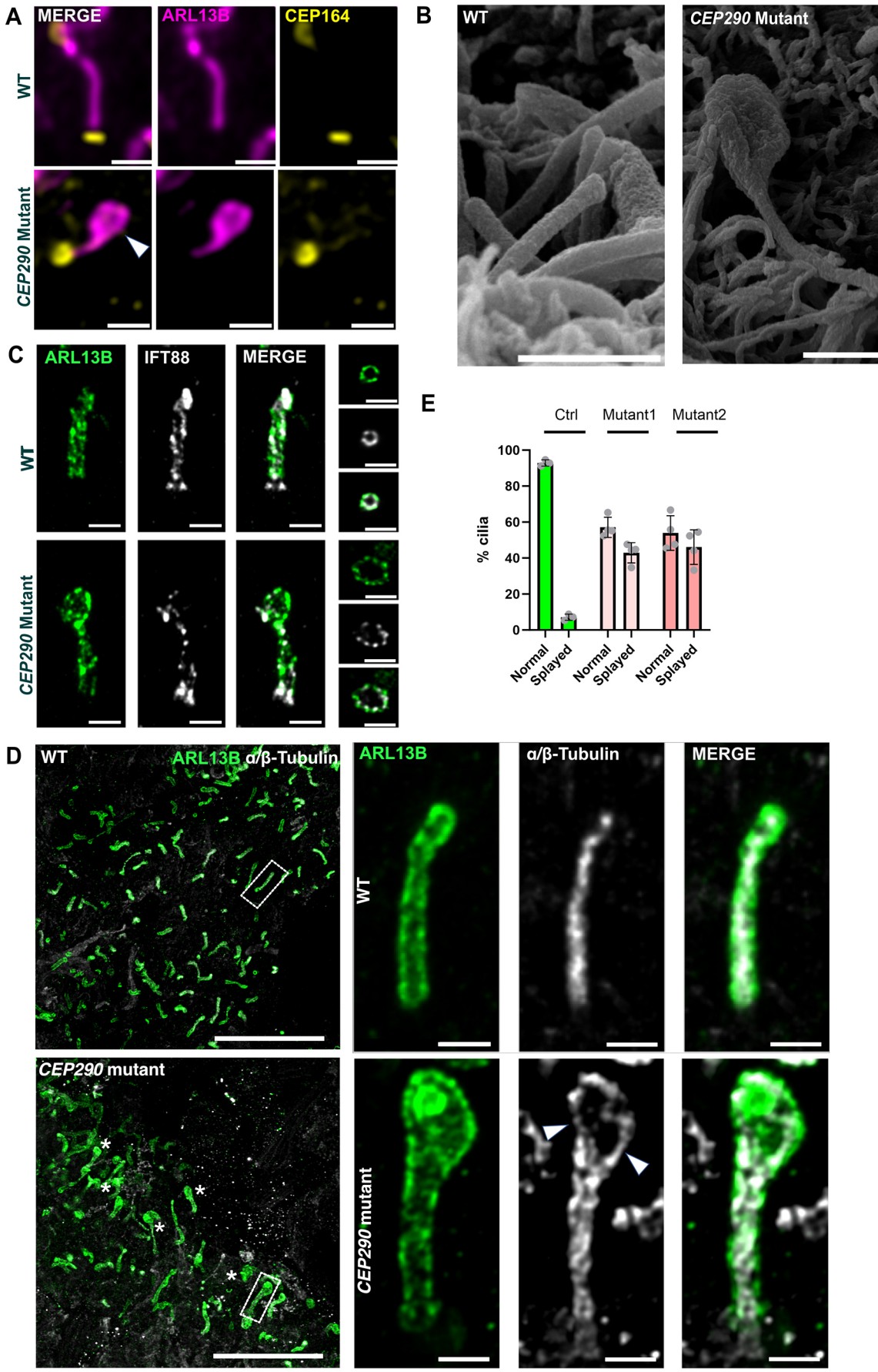

**Fig. 3.** See next page for legend.

**Fig. 3. Morphologically abnormal ventricular lumen cilia in *CEP290* mutant organoids.** (A) Immunohistochemistry on day 15 brain organoids focusing on cilia in the ventricular lumen stained with anti-CEP164 (basal body marker in yellow) and anti-ARL13B (ciliary membrane marker in purple). Scale bars: 2 µm. Note the racket-like shape of cilia in mutants with expansion of the apical ciliary tip region (arrowhead). (B) Scanning electron microscopy images of cilia in the ventricular lumen of 45-day-old WT and *CEP290* mutant organoids confirms the bulging of the apical ciliary region. Scale bars: 1 µm. (C,D) Immunohistochemistry with expansion super-resolution microscopy (UxEM) of cryosections of WT and *CEP290* mutant organoids at day 45. (C) Anti-ARL13B (green) and anti-IFT88 (white) staining shows IFT88 distribution along the cilium, with accumulation at the ciliary tip and base, in both WT and mutant cilia. Scale bars (post-expansion physical scale): 2.1 µm (corresponds to 500 nm original size, corrected for 4.2× expansion factor). (D) Overview images in UxEM with anti-ARL13B and anti-α/β-tubulin staining (left). The boxed area in WT and *CEP290* mutant overview image is enlarged (right). Asterisks highlight abnormal cilia. Ciliary membrane marker ARL13B (green) and axonemal MT marker α/β-tubulin (white) show characteristic cilia morphology in WT with organized α/β-tubulin signal marking the axonemal MT tightly sheathed by the ciliary membrane. *CEP290* mutant organoids present with various defects in MT integrity at the distal part of the axoneme (arrowheads). Scale bars (post-expansion physical scale): 30 µm (WT overview image); 2.1 µm (WT magnifications); 25 µm (*CEP290* mutant overview image); 2.1 µm (*CEP290* mutant magnifications). (E) Quantification of normal and abnormal (splayed MTs) cilia of overview images in UxEM. Graph shows mean ±s.d. (each datapoint represents one biological replicate, three biological replicates were quantified with a *n*>200 cilia per condition). Images in A are representative of at least three organoids from at least three independent differentiation runs. Images in C and D are representative of at least three organoids from at least two separate experiments.

An open question remains why only a subset of cells in these mutant brain organoids displayed abnormal cilia, which might be explained by incomplete loss-of-function due to nonsense-induced exon skipping, a phenomenon also observed in individuals with ciliopathies (Barny et al., 2018). This incomplete penetrance might also explain the relatively subtle defects observed in the differentiation of mutant organoids, and we cannot rule out that a complete knockout could result in more dramatic neuronal differentiation defects. However, the mild defects observed in our brain organoids are consistent with the relatively subtle morphological central nervous system (CNS) defects observed in most brain regions of individuals with pathogenic *CEP290* variants, thereby supporting the disease relevance of these brain organoid models.

The exact mechanism by which deficiency of a TZ protein like CEP290 leads to distal splaying of axonemal MTs remains to be determined. Comparable ciliary tip changes have been observed in brain ventricles of murine brains defective for *Tmem67*, another ciliary TZ protein (Abdelhamed et al., 2013). Our findings raise the question of what structural elements are responsible for maintaining the cohesion of MTs within the primary cilium, far away from the TZ. In motile cilia from *Tetrahymena*, a network of MT inner proteins weaves into the tubulin lattice and forms an inner sheath that maintains the cohesion of the MT doublets (Ichikawa et al., 2019). Whether such a lattice exists in primary cilia as well remains unexplored to date to our knowledge. Recent work using expansion microscopy in zebrafish and mouse embryos suggests that primary cilia do not display homogeneous MT length along the ciliary axoneme with the nine MT doublets not all reaching to the ciliary tip. Instead, the authors described axonemal thinning along the proximo-distal axis (Steib et al., 2023). Our work shows that WT cilia in the lumen of brain organoids display a relatively regular axonemal width and that the MT doublets mostly reach to the ciliary tip. The MT abnormalities observed in *CEP290* mutant cilia start as early as the middle region of the axoneme or concern only the tip region with an initially apparently conserved cohesion. One

hypothesis that could explain these findings is that the absence of a functional transition zone might impair the gating mechanism, leading to secondary lack of inner scaffold protein(s) through trafficking defects. Which proteins actually could act as such scaffolds remains to be determined in future work.

In summary, we identify striking ciliary morphological abnormalities within ventricular lumen of brain organoids. Such ciliary defects, characterized by loss of axonemal MT integrity and bulging ciliary tip are likely to be reflected in dysfunctional ciliary signaling which might contribute to the CNS abnormalities observed in individuals with *CEP290* pathogenic variants. Further work will be required to pinpoint the mechanism underlying the observed structural ciliary defects and the resulting signaling aberrations. Our study emphasizes the importance of examining disease-relevant cell types and highlights the utility of hiPSC-derived models that can show phenotypes not previously observed, even for an abundantly studied disease gene such as *CEP290*.

## MATERIALS AND METHODS
### Cell lines and culture conditions
The control hiPSC line HMGU1 was acquired from the Helmholtz Zentrum München, Germany. The generation of the HMGU1 isogenic lines: *CEP290* mutant hiPSC line 1 (p.L1749Sfs*10) and *CEP290* Mutant hiPSC line 2 (p.L1787Hfs*12) as well as CRISPR control line are described in Figueiro-Silva et al. (2025). Amniocytes were reprogrammed to iPSCs (patient clone C3) after informed consent from the parents and ethical approval from the local ethics commission (BASEC-Nr. 2019-00016) were obtained, in accordance with the ethical principles for medical research involving human subjects, outlined in the Helsinki declaration of 1975 (revised 2013). Amniotic fluid cells were reprogrammed by the iPSCore at the Institute for Regenerative Medicine, UZH using the CytoTune-iPSC 2.1 Sendai Reprogramming Kit (Thermo Fisher Scientific, #34546) as per the manufacturer's protocol.

From day 20, emerging colonies were manually selected based on morphology, then cultured and passaged as hiPSC. Clone C3 was selected and passed quality control as presented in Fig. S12 (Steeg et al., 2021).

All hiPSC lines used in this study were maintained at 37°C and 5% $CO_2$ atmosphere. In brief, feeder-free cultures were grown on Geltrex™-coated plates (Gibco, Thermo Fisher Scientific, #A1413302) in StemFlex™ (Gibco, #A3349401) and passaged using 0.5 mM EDTA (Invitrogen™, #15575020) in PBS.

Cultures were quality controlled for copy number variants by CGH-Array and routinely tested for mycoplasma using the VenoR® GeM Classic Mycoplasma detection kit (Minerva biolabs, #11-1100).

### Differentiation of hiPSCs into brain organoids
Brain organoids were generated using the protocol described previously (Lancaster et al., 2013; Lancaster and Knoblich, 2014), with a few modifications. The hiPSCs were passaged at least twice after thawing, ensuring the absence or low presence of differentiating cells and compact colony morphology. At day 0, hiPSCs were detached with PBS-EDTA, subsequently dissociated with Accutase (Gibco, Thermo Fisher Scientific, #A1110501). Single cells were plated at a density of 4500 cells per well in an ultra-low attachment 96-well plate (Corning® CAT#CLS7007-24EA); this is 50% less than described in the original Lancaster and Knoblich (2014) protocol. Cells were maintained in StemFlex medium supplemented with a low concentration (4 ng/µl) of basic fibroblast growth factor (bFGF; Peprotech, #100-18B-50uG) and 50 µM ROCK inhibitor Y-27632 (Lucerna Chem, #MCE-HY-10583-10MG). On day 2, medium was changed with, supplementing only bFGF. By day 4 of differentiation, embryoid bodies (EBs) reached a diameter of 350–600 µM and neural induction was initiated. The 96-well plates containing the EBs were washed twice with 200 µl neural induction medium (Lancaster and Knoblich, 2014) and cultured in 150 µl neural induction medium. Medium changes were performed every 2 days. After 4 days, the EBs are expected to form a primitive neuroepithelium, which showed by brightening smooth edges of the EBs, visible in light microscopy

using a 4× objective. Once the neuroepithelium was formed (usually day 8), each EB was embedded in 25 µl hESC-grade Matrigel (Corning®, #FAL354277) and stationary differentiation in 60 mm culture dishes (Corning®, #CLS430589-500EA) was started. Each 60 mm dish contained 16–20 EBs in 5 ml brain organoid differentiation medium without vitamin A (Lancaster and Knoblich, 2014). These stationary cultures were kept in the incubator for 4 days with a full medium change after 48 h. After the stationary culture phase, access Matrigel was removed and medium was changed to 5 ml brain organoid differentiation medium containing vitamin A (Lancaster and Knoblich, 2014). Brain organoids were differentiated up to day 80 on an orbital shaker at 37°C and 5% $CO_2$ (shaker orbit: 19 mm; shaking speed 74 rpm). Full medium changes were performed three times a week. All cell culture work was conducted in a biological safety cabinet under sterile conditions.

### Western blotting

Proteins were extracted from ∼70–80% confluent hiPSCs and 60-day-old brain organoids. Lysates were prepared using radioimmuoprecipitation assay (RIPA) lysis buffer. Protein quantification was performed using the Pierce™ BCA Protein Assay Kit (Thermo Fisher Scientific, #23225). Protein samples were denatured in 4× Laemmli buffer (Bio-Rad, #1610747) containing 2.5% β-mercaptoethanol at 95°C for 5 min. For each sample, 10 µg protein was loaded and separated on 4–12% Tris-Glycine Mini Gels (Bio-Rad, #4568093) Subsequently, proteins were transferred to polyvinylidene fluoride (PVDF) membranes (Bio-Rad, #1620174) by wet transfer following standard protocols. The following primary antibodies were used: 1:500 mouse anti-CEP290 (Santa Cruz, #sc390462), 1:2500 mouse anti-β-actin (Abcam, #49900) antibodies. Western blots were developed using 1:2500 anti-mouse-IgG secondary antibody conjugated with horseradish peroxidase (HRP) (Jackson Laboratories, #115-035-062) and chemiluminescent substrate Clarity™ MAX Western ECL (Bio-Rad, #1705061). Imaging was performed with ChemiDoc MP imaging system (Bio-Rad, #1708265) with ImageLab software.

### Histology and immunofluorescence of brain organoids

Organoids were washed in PBS and fixed with 4% (w/v) paraformaldehyde at room temperature (RT) with fixation times depending on the size and age of the organoids: 10 min for 15-day-old organoids, 15–20 min for 30–45-day-old organoids, and 30 min for 60–80-day-old organoids. Once the fixation period elapsed, organoids were quenched with 0.1 M PBS-Glycine for 5 min. This was followed by three 5-min washes using 1× PBS only.

In order to prepare for cryosectioning, samples were cryoprotected in 30% sucrose solution (in PBS) at 4°C overnight. Subsequently, organoids were embedded into OCT compound (Tissue Tek, #62550-01) and snap frozen with liquid nitrogen. Embedded frozen organoids were cryosectioned with a Cryostat (Cryostar NX50, Thermo Fisher Scientific) in 16 µm slices and sections were mounted on SuperFrost plus slides (Epredia™, #10149870).

To perform immunofluorescence staining, sections were thawed for 10 min, washed with PBS and blocked for 2 h with 5% BSA (Sigma, #A3294), 5% normal goat serum (NGS; Life Technologies, #31873) and 0.3% Triton-X-100 (Panreac, #A4975) in 1× PBS (Gibco, #10010023) blocking solution. Subsequently blocked sections were washed with 1× PBS once and incubated with primary antibodies diluted in 1% BSA, 1% NGS and 0.1% Triton-X-100 in 1× PBS at 4°C overnight. Sections were washed three times with 1× PBS and incubated with secondary antibodies diluted in 1% BSA, 1% NGS and 0.1% Triton-X-100 in 1× PBS for 2 h in dark at RT. Antibody-stained slides were washed at least three times with 1× PBS followed by a 10 min nuclei counterstain with NucBlue™ (Invitrogen™ #R37606). After a final wash with 1× PBS, slides were mounted with ProLong™ Gold Antifade mountant (Thermo Fisher, P36930).

Primary antibodies used were: 1:500 mouse anti-ARL13B [N295B/66] (Biolegend, #857601); 1:200 mouse anti-CEP164 (Santa Cruz Biotechnology, #sc-515403); 1:100 mouse anti-CEP290 (Santa Cruz Biotechnology, #sc-390462); 1:300 mouse anti-GFAP (Cell Signaling, #GA5); 1:300 mouse anti-MAP2A+B (IgG1) (Sigma, #MAB378); 1:200 mouse anti-S100b (Sigma, #S2532), 1:300 mouse anti-nestin (Millipore, #MAB5326); 1:500 rabbit anti-CEP290 (Abcam, #ab85728); 1:500 rabbit anti-ARL13B (Proteintech,

#17711-1-AP); 1:500 rabbit anti-GAD65 (Thermo Fisher Scientific, #PA5-85750); 1:300 rabbit anti-PAX6 (Biolegend, #Poly19013); 1:300 rabbit anti-SOX2 (Cell Signaling, #mAB3579); 1:200 rabbit anti-TBR1 (Abcam, #ab31940); 1:500 rat anti-CTIP2 (25B6) (Abcam, #ab18465); and 1:100 sheep anti-TTR (Prealbumin) (Bio-Rad ,#AHP1837).

Secondary antibodies used were: 1:1000 Alexa Fluor (AF) 488 goat anti-mouse-IgG (Life Technologies, #A21121); 1:1000 AF568 goat anti-mouse-IgG (Life Technologies, #A21124); 1:1000 AF568 goat anti-rat-IgG (Life Technologies, #A11077); 1:1000 AF647 goat anti-rabbit-IgG (Life Technologies, #A21245); and 1:1000 AF647 goat anti-sheep-IgG (Life Technologies, #A21448).

### Immunohistochemistry for adherent cells

hiPSCs were seeded in IBIDI eight-well chamber plates (IBIDI, #80821) and left to grow to 70–80% confluence. Medium was aspirated from the wells requiring fixation, followed by a single wash with 1× PBS. Subsequently we introduced 4% PFA and allowed the adherent cells to incubate at RT for 10 min. The cells were quenched with 0.1 M PBS-glycine for 5 min, and washed twice for 5-min using 1× PBS. Cells were permeabilized and blocked and stained in a similar manner to that described in the brain organoid staining protocol above with primary antibody incubation overnight at 4°C and secondary antibodies 2 h in dark at RT. After nuclei counterstain and a final wash with 1× PBS, wells were mounted with non-hardening IBIDI mounting medium (IBIDI, #50001).

Primary antibodies used were: 1:200 mouse anti-OCT3/4 (Santa Cruz Biotechnology, #sc-5279); 1:250 rabbit anti-Ki67 (Abcam, #Ab16667); and 1:500 rat anti-PH3 (Abcam, #ab10543).

Secondary antibodies used were: 1:1000 AF568 donkey anti-goat-IgG (Life Technologies, #A11057); 1:1000 AF488 goat anti-mouse-IgG (Life Technologies, #A21121); 1:1000 AF568 goat anti-mouse-IgG (Life Technologies, #A21124); 1:1000 AF568 goat anti-rat-IgG (Life Technologies, #A11077); and 1:1000 AF647 goat anti-rabbit-IgG (Life Technologies, #A21245).

### Confocal and widefield microscopy

Confocal images were acquired using the laser-scanning microscope (CLSM Leica Sp8-inverse). The microscope was equipped with a HC Pl APO CS2 63×/1.4 oil objective. Widefield imaging was undertaken on a Zeiss Axio Scan Z1. Images were taken with Plan Apochromat objectives with either 10 or 20× magnification. The microscope was equipped with a Hamamatsu Orca Flash 4.0 monochrome camera 16bit (fluorescence camera). Fluorescence filters for following fluorophores were used: DAPI/Hoechst 33342 [excitation 385, bandpass filter (BP) 450/40], eGFP (excitation 436/20, BP 525/50), Cy3 (excitation 555, BP 605/70) and Cy5 (excitation 630, BP 690/50). Confocal and widefield imaging was performed at the Center for Microscopy and Image Analysis core facility of the University of Zurich (ZMB core facility).

### Ultrastructure expansion microscopy

U-ExM was performed on fixed and sectioned brain organoids. The method of expansion microscopy was adapted from the U-ExM method by Gambarotto et al. (Arsenijevic et al., 2024; Gambarotto et al., 2021). Briefly, frozen sections on slides were thawed at RT for 2 min. Then, a double-sided sticky spacer of 0.3-mm thickness (IS317, SunJin Lab Co.) was stuck on the slide around the section of interest. Then, the crosslinking prevention step was performed by incubation of 2% acrylamide (AA; A4058, Sigma-Aldrich) and 1.4% formaldehyde (FA; F8775, Sigma-Aldrich) in a total volume of 300 µl at 37°C for 3 h. After removing the first solution, the gelation step was done by adding 130 µl of monomer solution composed of 75 µl of sodium acrylate [stock solution at 38% (w/w) diluted with nuclease-free water, 408220, Sigma-Aldrich], 37.5 µl of AA, 7.5 µl of N,N′-methylenbisacrylamide (BIS, 2%, M1533, Sigma-Aldrich), 10 µl of 10× PBS together with ammonium persulfate (APS, 17874, Thermo Fisher Scientific) and tetramethylethylenediamine (TEMED, 17919, Thermo Fisher Scientific) as a final concentration of 0.5%, for 1 h and 30 min at 37°C. Of note, APS and TEMED have to be added at the last minute to avoid premature polymerization. A 24-mm coverslip was added on top to close the chamber. Next, the coverslip and the spacer were removed, and the slide was submerged in a 50 ml Falcon tube filled with denaturation buffer

pre-heated at 95°C [200 mM SDS, 200 mM NaCl, 50 mM Tris base in water (pH 9)]. The Falcon tube was then incubated for 2 h at 95°C in a water bath. Finally, the gel was rinsed and expanded in three successive ddH$_2$O baths, before processing to the immunostainings. Immunostainings were performed by first incubating expanded gels in three PBS 1× baths of 5 min. Subsequently, primary antibodies in PBS with 2% of bovine-serum album (BSA) were added overnight at 4°C: anti-IFT88 (Proteintech #13967-1-AP), 1:250; anti-ARL13B (Proteintech #17711-1-AP), 1:250; anti-α-tubulin (ABCD antibodies #ABCD_AA345), 1:250; and anti-β-tubulin (ABCD antibodies #ABCD_AA344), 1:250. After three additional PBS with 0.1% Tween 20 (PBS-T) washes for 5 min each, secondary antibodies were incubated for 3 h at 37°C. After final three washes in PBS-T, expanded gels were mounted onto 24 mm poly-D-lysine (01 mg/ml)-coated coverslips. Images were acquired with an inverted Stellaris 8 microscope. Images were taken with a HC Pl APO CS2 63×/1.4 oil objective with 5× optical zoom at maximum resolution, adaptive as 'Strategy' and water as 'Mounting medium' to generate deconvolved images. 3D stacks were acquired with 0.27 µm z-intervals and a 36 nm x,y pixel size.

### Scanning electron microscopy
Brain organoids were washed once with 1× PBS and fixed in SEM fixative (0.1 M cacodylate, 2.5% gluteraldehyde) for 35 min at RT. Dehydration and mounting steps were performed following standard protocols. Images were obtained on a Zeiss Supra VP 50 microscope.

### Cilia imaging and quantification
Brain organoid-sections stained for ciliary marker (ARL13B as axonemal marker, CEP164 as basal body marker) were imaged using a Leica SP8 (confocal microscope at 63× with 7.5 optical zoom). Images were acquired every 0.2 µM in the z-plane. Z-stacks from at least three ventricles of three different organoids in three independent organoid differentiation were captured. Performed imaging and analysis of ventricular zone projected cilia took place at different time points of differentiation (day 15, 30 and 45).

Images were deconvoluted using Huygens software and subsequently analyzed using the ImageJ/Fiji plug in CiliaQ (Hansen et al., 2021). The channel for ciliary length measurement (ARL13B staining) was used for segmentation and subsequent length assessment. The segmentation method ReyniEntropy was applied; with an additional blur with Gaussian at 1.5. The unsegmented ARL13B channel was used to assess fluorescence intensity by applying the mask generated in the segmented channel. Minimum cilia size was set to 10, and basal body stain channel (CEP164) was used for orientation detection of the ciliary base/tip. Cilia touching x/y/z borders were excluded.

### Cortical unit imaging and quantification
To quantify cortical unit development, the expression pattern of TBR1, a post mitotic neuronal marker indicative of the developing preplate, and of CTIP2, a marker for deeper cortical layers V and VI at day 80 was assessed. Immunolabeling was employed to analyze cortical development in brain organoids, and relative thickness was quantified. Ventricular zone, 'layer 1' (mostly representing TBR1$^+$ cells) and 'layer 2' (mixed population of TBR1$^+$ and CTIP2$^+$ cells) were defined. The relative thickness of these layers was measured and averaged along three lines set apart by 45° angles. Data was obtained from three independent differentiations (N=3), for each differentiation three cortical structures from three organoids were assessed.

### scRNA-seq of brain organoids
For each condition, three whole brain organoids were pooled and washed three times in HBSS. Dissociation was performed using the gentleMACS™ Dissociator with the Neural Tissue Dissociation Kit in C tubes following the manual's instructions. After running all the dissociation steps on the gentleMACS Dissociator and in between rotation steps using the MACS mix tube rotator at 37°C, the enzyme mix was stopped by adding 5 ml of HBSS+5% FBS. The cell solution was centrifuged for 5 min at 300 $g$ and resuspend in 5 ml HBSS plus 5% FBS. First, the cell suspension was passed through a 40 µm cell strainer and secondly through a FACS tube strainer. Finally, an ultra-slow centrifugation step at 90 $g$ for 10 min was added

to eliminate gross cell debris. Samples were sequenced paired-end on an Illumina Novaseq 6000 instrument at the Functional Genomics Center Zurich.

### Single-cell RNA-seq analysis
Data analysis was performed using the Seurat package (version 4.3.0.1) (Hao et al., 2024). Low-quality cells were filtered out based on library size, number of detected genes, doublets and percentage of mitochondrial reads, resulting in 8281 and 10,145 high-quality cells remaining per sample. Normalization was performed using the SCTransform method, which estimates variance from the filtered data and identifies the most variable genes for each sample independently. The datasets were merged without batch correction, as no differences between batches were identified and to maintain possible differences in cell types between conditions. Next, UMAP calculation and clustering were performed based on 15 principal components with a resolution of 0.22. For cluster marker identification using Wilcoxon rank sum tests, a minimal log fold change of 0.25 and a minimal expression in 25% of the cells in the specified cluster were set as threshold. Cell type identification was performed with enrichR using PanglaoDB 2021 database (Franzén et al., 2019) combined with validation of a set of common expression markers. To further study the choroid plexus cluster, we ran Wilcoxon rank sum tests for several sets of genes for mutant choroid plexus cells versus all other mutant cells for genes expressed in at least 25% of cells in either of the two groups. We also compared the expression of the same sets of genes between mutant and WT cells in the choroid plexus cluster using Wilcoxon rank sum tests. Finally, we performed differential expression analysis between mutant and WT cells using MAST (Finak et al., 2015). Using the fold changes of this model, we ranked the genes and performed gene set enrichment analysis with 100,000 permutations.

### Statistical analysis
Statistical analysis was performed on GraphPad Prism (GraphPad version 9.5.1). We tested every dataset for normality distribution by performing a Shapiro–Wilk test. If normality passed, datasets were analyzed by two-way ANOVA. If data was not normally distributed, non-parametric statistical analysis with Kruskal–Wallis was performed. $N$ indicates independent biological replicates from distinct samples. Data are all represented as scatter dot plots with error bars representing mean±s.d. Differences with values of $P<0.05$ were considered statistically significant (*$P<0.05$, **$P<0.01$, ***$P<0.001$).

### Acknowledgements
We acknowledge the technical support of the Core Facility iPSC at Helmholtz Zentrum München which provided the HMGU1 hiPSC line.

### Competing interests
The authors declare no competing or financial interests.

### Author contributions
Conceptualization: M.E., O.M., R.B.-G.; Data curation: M.E., E.M.A.; Formal analysis: M.E., O.M., E.M.A., L.P., J.F.-S., A.A., M.M.; Funding acquisition: A.R., P.G., V.H., R.B.-G.; Investigation: M.E., O.M., E.M.A., L.P., J.F.-S., A.A., M.M., M.G.; Methodology: M.E., O.M., E.M.A., A.A., M.G., R.B.-G.; Project administration: R.B.-G.; Resources: M.G., A.R., P.G., V.H., R.B.-G.; Supervision: J.F.-S., A.A., P.G., V.H., R.B.-G.; Visualization: M.E., O.M., E.M.A., L.P.; Writing – original draft: M.E., O.M., R.B.-G.; Writing – review & editing: M.E., O.M., E.M.A., L.P., J.F.-S., A.A., M.M., M.G., A.R., P.G., V.H., R.B.-G.

### Funding
This work was supported by the Clinical Research Priority Program of the Medical Faculty of the University of Zurich Praeclare, by the University Research Priority Program of the University of Zurich AdaBD, by the Swiss National Science Foundation grants PP00P3_198895 and 310030_22012 to R.B.-G and by a grant from the Uniscientia foundation (UZH foundation) to R.B.-G. Open Access funding provided by University of Zurich. Deposited in PMC for immediate release.

### Data and resource availability
All relevant data and details of resources can be found within the article and its supplementary information. All unique and stable reagents generated in this study are available from the lead contact with a completed materials transfer agreement. Raw and processed scRNA-seq data has been deposited in the Gene Expression Omnibus (GEO accession no. GSE293717).

## Peer review history
The peer review history is available online at https://journals.biologists.com/jcs/lookup/doi/10.1242/jcs.264092.reviewer-comments.pdf

## Special Issue
This article is part of the Special Issue 'Cilia and Flagella: from Basic Biology to Disease', guest edited by Pleasantine Mill and Lotte Pedersen. See related articles at https://journals.biologists.com/jcs/issue/138/20.

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
