## [Peer Review File · Journal of Cell Science]

CEP290 deficiency disrupts ciliary axonemal architecture in human iPSC-derived brain organoids

Melanie Eschment, Olivier Mercey, Ellen M. Aarts, Ludovico Perego, Joana Figueiro-Silva, Michelle Menzel, Affef Abidi, Melanie Generali, Anita Rauch, Paul Guichard, Virginie Hamel and Ruxandra Bachmann-Gagescu

DOI: 10.1242/jcs.264092

Editor: Pleasantine Mill

Review timeline

Original submission:	17 April 2025
Editorial decision:	27 May 2025
First revision received:	28 August 2025
Accepted:	19 September 2025

Original submission

First decision letter

MS ID#: jcs.264092

MS TITLE: CEP290-deficiency disrupts ciliary axonemal architecture in human iPSC-derived cerebral organoids

AUTHORS: Melanie Eschment; Olivier Mercey; Ellen Aarts; Ludovico Perego; Joana Figueiro-Silva; Affef Abidi; Melanie Generali; Paul Guichard; Virginie Hamel; Ruxandra Bachmann-Gagescu

ARTICLE TYPE: Short Report

Dear Ruxandra and team,

Thank you for submitting to our Special Issue on Cilia and Flagella for the Journal of Cell Science. I am pleased to share that we have now reached a decision on the above manuscript.

To see the reviewers' reports and a copy of this decision letter, please go to:

As you will see, the reviewers gave favourable reports but raised some points that will require amendments to your manuscript. I hope that you will be able to carry these out because I would like to be able to accept your paper, depending on further comments from reviewers.

Overall, a high-quality short report of two patient CEP290 iPSCs leveraging cutting-edge scRNA sequencing and expansion super-resolution microscopy for a primarily descriptive study of corticogenesis in organoids. Additional information characterizing CEP290 mRNA and protein in cerebral organoids is asked by all three reviewers, plus more characterization of the racket phenotype and discussion on propensity to form CP lineages, as well as broader discussion on specific neurodevelopmental conditions and neural cilia defects.

Reviewer 1

Advance summary and potential significance to field

Eschment et al. have explored the role of CEP290 mutations in human neuronal tissue development using two lines of human iPSC-derived cerebral organoids as a model. The authors show that CEP290 loss, from putative null mutations in CEP290 exon 39, do not affect cell cycle progression or overall cerebral organoid formation with only minor effects on neuronal differentiation. However, CEP290 mutations do cause a distinctive ciliary phenotype characterised by bulging ciliary membranes and splayed distal axonemal microtubules. This defect in ciliary microtubule architecture suggests new avenues to understand the pathomechanisms that lead to CNS abnormalities in CEP290-related ciliopathies such as Joubert syndrome.

Comments for the author

Although the short report has data of high technical standard, comprising scRNA sequencing and expansion super-resolution microscopy, this is essentially a descriptive study. There is some important information missing from this manuscript:

1. Confusingly, the two mutant iPSC lines have already been characterized by the same group in a separate biorxiv pre-print: doi: <https://doi.org/10.1101/2025.03.31.646311> (cited as Figueiro-Silva et al. 2025 in the manuscript). In the pre-print, the authors demonstrate partial skipping of exon 39, in addition to lower levels of CEP290 RNA and protein. Since, iPSC cells and differentiated cells do not always behave similarly in terms of gene expression, the authors should examine CEP290 RNA expression levels and exon-skipping in the cerebral organoids. Presumably, this information is easily available from the RNA seq data and can be confirmed by rt-PCR.
2. The authors need to show CEP290 ciliary localization, mislocalization or absence using IF microscopy in iPSCs and cerebral organoids. In particular, it is important to see if the difference between normal and "racket-shaped" cilia in mutant organoid correlates with aberrant CEP290 localization. (I should add that IF staining of CEP290 in organoid cryosections is tricky, but we have an optimised protocol adapted from Mike Cheetham's lab that works well. The key step is not to fix organoids before embedding and freezing in OCT before cryosectioning.)
3. What do cilia look like in mutant iPSC cells? Are they normal looking or racket-like? Does the ciliary morphology defect only arise in organoids? This phenotype is not unique and has been observed in other iPSC-derived models, for example RPE and retinal organoids, in which ciliogenesis is defective (see Buskin et al. Nat Commun 2018;9:4234) and the differentiated cells are under stress.
4. Are the defective organoid cilia of normal length? Figure S6 only measures the length of the normal-looking cilia and does not report data about "racket-shaped" cilia. Length aberrations of defective cilia would suggest defects in ciliary tip module functioning. Usually, lack of CEP290 reduces cilia incidence. Is the percentage of total cilia decreased in mutants or is it similar to wildtype?
5. The scRNAseq data is very interesting. Would the authors provide GO terms enrichment for Molecular Function (MF), Cellular Component (CC), and Biological Process (BP) to shed more light on the signalling pathways. This might shed light on the tendency of mutants to form choroid plexus cells.
6. Figure 3C appears to show more IFT88 at the wildtype ciliary tip compared to the mutant, and more at the ciliary base of the mutant compared to the wildtype. However, this figure by itself does not rule out IFT defects, although it suggests that IFT88 is not accumulating at the mutant tip. (A panel of 3 or 4 representative cilia images could be included to document the range of phenotypes). Could the authors discuss why the cilia length and/or IFT is unaffected, despite the splaying of the ciliary axonemal tips? Do they suspect that the ciliary tip module is unaffected?
7. If IFT is normal, what is causing the "racket-like" ciliary tip phenotype? The authors speculate that this is due to a lack of inner scaffold proteins. Can this be shown with expansion microscopy, which would be an approach to gain some initial mechanistic insight. Is this a phenotype that is specific to cerebral organoids and not to other cell types (assuming that mutant iPSCs do not have this ciliary defect)?

Minor comments:

1. Line 55 should be Figure 3C not 3D
2. Wildtype control panel needed for Figure 2E
3. A schematic showing the location of mutations (CRISPR and amniocyte-derived) on a map of the CEP290 gene and exon structure would be useful

Reviewer 2*Advance summary and potential significance to field*

Eschment et al. generated CEP290-deficient cerebral organoids using both CRISPR/Cas9-edited CEP290 KO iPS cells and patient-derived iPS cells. Loss of CEP290 did not affect the cell cycle in iPS cells. While CEP290 deletion did not affect neurogenesis during cerebral organoid formation, CEP290 deletion increased the number of organoids with choroid plexus. The authors also observed cilia with splayed morphology in the ventricles of cerebral organoids. These data are interesting to understand the function of CEP290 in ciliogenesis and brain development.

Major Points:

1. The authors showed that 11 out of 43 CEP290 mutant organoids developed choroid plexus, compared to 1 out of 39 control organoids. It is interesting that CEP290 deletion enhanced choroid plexus generation. The authors may include a graph about choroid plexus generation and TTR-stained images of choroid plexus from control cerebral organoids in Figure 2. However, Lancaster et al. (2013) reported the existence of choroid plexus in their cerebral organoids (25 out of 35), whereas only 1 out of 39 control organoids in this study. This discrepancy should be discussed.
2. Did the choroid plexus contain cilia with abnormal morphology in the CEP290 mutant organoids?
3. The authors mentioned that Wnt signaling is not changed in the choroid plexus cluster based on single cell RNA-seq data. Can the authors show the Wnt signaling data of the choroid plexus cluster in the supplementary figure? Were other signaling pathways such as Shh, BMP, or mTOR differentially expressed in this cluster? I understand that single cell RNA-seq data is not easy to interpret when the number of cells in the cluster is too low. If the authors think it is appropriate not to show the data to avoid confusing readers, that is understandable.
4. Could the authors perform a western blot of CEP290 on CEP290 KO iPS cells?
5. In Figure S6, the authors showed GFAP-positive cells. However, the image quality is too low to determine whether these cells are astrocytes or background. I want to see high-mag images. In addition, are these GFAP-positive cells also positive for S100B?

Minor Comments

1. Please include the catalogue numbers of reagents in the Methods section.
2. Western blot protocol is not written in the Methods section.
3. Nanog is listed in the Methods section and the legend of supplementary Figure 7, but no Nanog data were shown.
4. In the ultrastructure expansion microscope experiments, when and how was immunostaining performed? Was the protocol, such as antibody concentration and incubation condition, similar to those used for "Histology and IF of cerebral organoids?"
5. In Figure 1A, asterisks are hiding the ventricles in the organoids. Please use arrows instead of asterisks.
6. In Figures 1C and 1D, it's difficult to understand which regions were stained. Please include high-mag images of the ventricular zones and the cortical plate zones, respectively.
7. In Figure 2A, please specify whether the data were from day 79 or day 80.
8. In Figure 2C, I think the mutant bar is narrower than the WT bar.
9. In the legend for Supplementary Figure 4, TBR1 is labeled as a subventricular zone (SVZ) marker.
10. In Supplementary Figure 4, please provide high-mag images of TBR1 and CTIP2 staining to assess the co-localization.
11. In Supplementary Figure 5A, CEP164 and ARL13B are difficult to distinguish due to the red and magenta colors. Please use a different color combination.
12. In Supplementary Figures 7D and 7E, please provide more explanation. What do "C+" and "C-" mean? Which band is CEP290 in 7E? What is A45468? What does clone C3 refer to?
13. In Supplementary Figure 8F, there is a stray straight line on the left side of the "normal" bar.

14. In the Ethics statement, the protocol number for human research approval was not provided. If available, please include it.

Reviewer 3

Comments for the author

Eschment et al, in this work, have generated human brain organoids and studied a previously unexplored area of cilia abnormalities in brain organoids due to mutations in *Cep290*, a ciliary gene implicated in TZ formation for proper ciliary function. In this aspect, the work has shown that the cilia are bulged at the distal tip, possibly accumulating IFT proteins, which could cause defects in ciliary functionality. It is a straightforward and descriptive work that will be helpful to the cilia community. The cilia data presented are impressive and robust. The following points should be carefully addressed before publication. Addressing these comments will enhance the manuscript and make it more attractive to the community.

In the abstract, it reads that ciliopathies are often connected to brain disorders, a sentence that is not entirely true. On the other hand, there are cilia mutations leading to neurodevelopmental disorders (although centrioles are structurally and numerically normal). As the paper addresses brain organoids that indicate neurodevelopmental disorders, it is worthwhile to discuss "non-syndromic" ciliopathies, which encompass brain developmental disorders that do not affect other organs. As more evidence emerges, it is critical to emphasize the aspect of neurodevelopmental disorders due to ciliary dysfunctions as non-syndromic ciliopathies. As the work majorly deals with defects in neural development in organoids differentiated in an undirected manner, call them brain organoids.

In which aspect of the organoid protocol has been changed? Clarify it. The scheme shown is confusing, as if *SOX2*, *Nestin*, and other NPC markers are provided to the culture at a specified period. I suggest removing the scheme and providing any details in text.

The authors have chosen undirected differentiation with the reasoning that directed differentiation may cause defects affecting primary cilia-mediated signaling. While it is intriguing, is there any citation that proves these? Cite them.

Figure 1A: Day 8 organoid cultures are referred to as EBs. Clarify this. EBs do not show neuroepithelium, and there is no characterization of the EBs that has been presented (ectoderm, mesoderm, and endoderm). Clarify whether they are day 8 organoids or EBs. It appears they are day 8 cultures. EBs appear well before differentiation is induced. Additionally, most panels display only one object, which raises concerns about the statistical power and robustness of the data.

Display a group of organoids or EBs in one panel.

FACS analysis of cell cycle stages is an interesting approach. I am curious about the analysis and which surface marker was used to sort fluorescently? Unless it is a flow cytometry.

Figure 1, showing the organoid morphologies, is unfortunately of low resolution and does not reveal cytoarchitectures. This is often the case with methods adapted from reagents related to Stem Cell technology. In this manner, I fail to see apicobasal polarity between NPCs and cortical neurons. Show a particular region (cropped with high magnification) to solve this uncertainty. The same is true for *GAD65* and *GFAP*. Show it if not; remove these panes, as *GFAP* is not specific. At the same time, *TBR1* and *CTIP2* stainings are very good. I would recommend that they adapt a zoomed-in image of a particular region and incorporate it into the central figure.

The Western blot for validating the clones in Figure S7 lacks the bands in one blot, the loading control, and appears to be a compilation of a few blots. This is crucial to address to prevent severe problems that may arise in the future. All iPSC lines (used to generate brain organoids, including HMGU1 lines) should be validated to demonstrate that they have *Cep290* deficiency.

There is no such thing as ventricular cilia. Ventricular Zones (again, this must be specified in the figures as VZ) is a place where neurogenesis begins. Apically, it forms a lumen called the ventricular lumen. Cells emanate primary cilia into this lumen. Ventricular lumen is far away from the cortical plate and cannot be named as the cortical lumen. Correct these terminologies. The paper overall lacks validation for iPSCs (in terms of showing pluripotency markers) and quantification. Provide quantifications for the VZ thickness and the primitive cortical plates. scRNA sequencing is a low-resolution experiment and, of course, will not reflect defects or misregulated components related to signaling. However, since the data is there, the author takes

all signalling components and tests their distribution across various cell clusters, which may reveal localized signals in the Choroid plexus cluster. Additionally, it will be crucial to display the cilia component distribution across various clusters and present them as quantitative dot plots. In this respect, the authors can consider Cep290 interacting components and genuine ciliary molecules.

First revision

Author response to reviewers' comments

Point-by-point response to reviewers

We would like to thank all reviewers and editors for their careful evaluation of our work and are grateful for their constructive comments which we have addressed as well as we could, resulting in an improved manuscript which we hope will satisfy all requests and allow publication.

Reviewer 1: SUMMARY OF THE ADVANCE MADE IN THIS PAPER AND ITS POTENTIAL SIGNIFICANCE TO THE FIELD

Eschment et al. have explored the role of CEP290 mutations in human neuronal tissue development using two lines of human iPSC-derived cerebral organoids as a model. The authors show that CEP290 loss, from putative null mutations in CEP290 exon 39, do not affect cell cycle progression or overall cerebral organoid formation with only minor effects on neuronal differentiation. However, CEP290 mutations do cause a distinctive ciliary phenotype characterized by bulging ciliary membranes and splayed distal axonemal microtubules. This defect in ciliary microtubule architecture suggests new avenues to understand the pathomechanisms that lead to CNS abnormalities in CEP290-related ciliopathies such as Joubert syndrome.

SUGGESTIONS TO AUTHORS

Although the short report has data of high technical standard, comprising scRNA sequencing and expansion super-resolution microscopy, this is essentially a descriptive study. There is some important information missing from this manuscript:

1. Confusingly, the two mutant iPSC lines have already been characterized by the same group in a separate bioRxiv pre-print: doi: <https://doi.org/10.1101/2025.03.31.646311> (cited as Figueiro-Silva et al. 2025 in the manuscript). In the pre-print, the authors demonstrate partial skipping of exon 39, in addition to lower levels of CEP290 RNA and protein. Since, iPSC cells and differentiated cells do not always behave similarly in terms of gene expression, the authors should examine CEP290 RNA expression levels and exon-skipping in the cerebral organoids. Presumably, this information is easily available from the RNA seq data and can be confirmed by rt-PCR.

- *The detailed description of the iPSC lines, including all quality controls, is now published as a resource paper in Stem Cell Research (Figueiro-Silva et al, <https://doi.10.1016/j.scr.2025.103781>). We have now added to the current manuscript additional characterization of the effect of the mutations in both lines with RT-PCR and WB in the organoids, which should hopefully clarify things for the readers.*

2. The authors need to show CEP290 ciliary localization, mislocalization or absence using IF microscopy in iPSCs and cerebral organoids. In particular, it is important to see if the difference between normal and "racket-shaped" cilia in mutant organoid correlates with aberrant CEP290 localization. (I should add that IF staining of CEP290 in organoid cryosections is tricky, but we have an optimised protocol adapted from Mike Cheetham's lab that works well. The key step is not to fix organoids before embedding and freezing in OCT before cryosectioning.)

- *We agree with the reviewer that it would be important to determine whether any residual CEP290 protein is mislocalized and how this would correlate with the abnormal ciliary morphology. Unfortunately, we have not been able to generate reliable stainings with either of two different CEP290 antibodies tried in any of the conditions we have tested, including unfixed organoids embedded before cryosectioning as suggested by the reviewer. We would also like to point out that despite the presence of CEP290 mRNA with*

skipped exon 39, the western blot shows almost no band in either mutant, and we believe that if there is any residual CEP290 protein lacking exon 39 produced, it would be at minimal levels, so we would predict that there should not be a correlation between (mis-)localization of residual CEP290 protein and ciliary phenotype.

3. What do cilia look like in mutant iPSC cells? Are they normal looking or racket-like? Does the ciliary morphology defect only arise in organoids? This phenotype is not unique and has been observed in other iPSC-derived models, for example RPE and retinal organoids, in which ciliogenesis is defective (see Buskin et al. Nat Commun 2018;9:4234) and the differentiated cells are under stress.

- *The reviewer poses an interesting question: we do not see such ciliary abnormalities in iPSCs and have added a supplementary figure showing cilia in iPSCs. Please note however that the cilia in iPSCs are really very short (at least with these lines) which could explain this difference (or which may mask morphological abnormalities).*
- *We agree that this is not the first description of similar morphological abnormalities, as we also cite other previous reports of similar abnormal cilia in the manuscript (Abdelhamed et al. 2013). Since the defects described in Buskin et al concern photoreceptors, which have such very highly specialized ciliary compartments, we feel that the comparison with the defects in the organoids could raise criticism and therefore would prefer not to additionally cite this work.*

4. Are the defective organoid cilia of normal length? Figure S6 only measures the length of the normal- looking cilia and does not report data about "racket-shaped" cilia. Length aberrations of defective cilia would suggest defects in ciliary tip module functioning. Usually, lack of CEP290 reduces cilia incidence. Is the percentage of total cilia decreased in mutants or is it similar to wildtype?

- *We have made the conscious decision not to measure the length of the abnormal cilia as we felt that it would be very hard to perform such a quantification in a fair and unbiased way: indeed, such a measurement cannot be performed systematically in an automated manner given the variable shapes observed and even a manual measurement may bias the results depending on which part of the bulging tip is chosen to terminate the measure. Moreover, we are not convinced that even a shortened ciliary length would provide strong evidence about the underlying mechanism (could still be defective ciliogenesis, axonemal instability, tip module defects or other mechanisms). Overall, the bulging cilia were certainly not remarkably shorter than the normal rod-shaped cilia as is visible in figure 3 and in supplementary figures S9, S14 and S15, which are representative images of what we observe. Concerning the ciliation rate, it is a very hard thing to accurately quantify in these ventricular lumen, as we cannot determine which cell any cilium belongs to (so cilia to nuclei ratio is not feasible). We could at most calculate the proportion of basal bodies that have Arl13b signal, but this also has limitations, for example since dividing cells will have a centrosome but no cilium. Overall, we did not see qualitatively any difference in ciliation rate as is shown in supplementary figure S9, which shows representative images.*

5. The scRNAseq data is very interesting. Would the authors provide GO terms enrichment for Molecular Function (MF), Cellular Component (CC), and Biological Process (BP) to shed more light on the signalling pathways. This might shed light on the tendency of mutants to form choroid plexus cells.

- *The analysis of the scRNAseq data has unfortunately not provided any very useful hints about signaling pathways affected by loss of CEP290 to explain the choroid plexus formation and we feel that adding a figure in the manuscript might lead to overinterpretation of the results. We have slightly expanded the results section on the scRNAseq data and added supplementary xls spreadsheets with the different analyses we have performed (in addition to uploading the raw data for data reuse), if readers are interested in seeing the results.*
- *Concretely, we have performed gene set enrichment analyses on the fold changes from mutant vs. wild-type cells and plotted the top 5 most significantly enriched GO terms (both up and downregulated in mutants - see plots below). Among these GO terms we find terms related to neuronal differentiation (for example "neuron to neuron synapse"), possibly explained by the slightly decreased proportion of more*

differentiated neuronal cells in mutants that we observe. Please note however that many GO terms are only driven by a very small number of genes.

- The majority of the terms (and associated genes) are not very contributive, such as those linked to ribosomes or extracellular matrix, and likely reflect secondary changes rather than pathways primarily driving the phenotype (see figure below and Supplementary Tables S3-S4).
- For a more targeted analysis of various signaling pathways and by cluster, please see our responses to reviewers 2 and 3 below.

6. Figure 3C appears to show more IFT88 at the wildtype ciliary tip compared to the mutant, and more at the ciliary base of the mutant compared to the wildtype. However, this figure by itself does not rule out IFT defects, although it suggests that IFT88 is not accumulating at the mutant tip. (A panel of 3 or 4 representative cilia images could be included to document the range of phenotypes). Could the authors discuss why the cilia length and/or IFT is unaffected, despite the splaying of the ciliary axonemal tips? Do they suspect that the ciliary tip module is unaffected?

- *This is a very good point. We have added a supplementary figure S14 with a gallery of IFT stainings in controls and mutants and have also added some discussion on why cilia length and IFT appear not to be affected in the main MS. Concerning the ciliary tip module, we had tried staining for CEP104 in the expanded samples but unfortunately could not get the antibody to work. Hence, we unfortunately cannot not make any statements about the ciliary tip module.*

7. If IFT is normal, what is causing the "racket-like" ciliary tip phenotype? The authors speculate that this is due to a lack of inner scaffold proteins. Can this be shown with expansion microscopy, which would be an approach to gain some initial mechanistic insight. Is this a phenotype that is specific to cerebral organoids and not to other cell types (assuming that mutant iPSCs do not have this ciliary defect)?

- *As discussed in the manuscript, the underlying mechanism explaining this morphological ciliary phenotype remains to be solved. Expansion microscopy could help answer this question if we knew what antibodies to use. As explained above, we tried to test the hypothesis that lack of the ciliary tip module may underlie this splaying of microtubules but could not get CEP104 staining to work and have no other antibodies targeting this complex that work. We hypothesize some sort of inner scaffold but to our knowledge, no data exists describing such a scaffold in primary cilia along the axoneme. Indeed, while IFT and axonemal structure and associated proteins have been thoroughly studied in Chlamydomonas and to some extent in the highly specialized primary cilium of the photoreceptors, much less data exists for primary cilia. Beyond comparison to iPSCs, we cannot make claims as to whether this phenotype is only found in cerebral organoids.*

Minor comments:

- Line 55 should be Figure 3C not 3D
 - *Corrected, thank you for noticing this typo*
- Wildtype control panel needed for Figure 2E
 - *We had not added a control image with TTR staining since we have not been able to generate one. Indeed, given the quasi-absence of choroid-plexus-like*

structures in controls, all slides that we stained for TTR were lacking these. We also do not necessarily expect such structures to be different in mutants compared to controls (based on published images from Lancaster et al), but just to occur more frequently in the mutants compared to their isogenic controls.

3. A schematic showing the location of mutations (CRISPR and amniocyte-derived) on a map of the CEP290 gene and exon structure would be useful
 - *Has been added in supplementary figure S1*

Reviewer 2: Eschment et al. generated CEP290-deficient cerebral organoids using both CRISPR/Cas9-edited CEP290 KO iPS cells and patient-derived iPS cells. Loss of CEP290 did not affect the cell cycle in iPS cells. While CEP290 deletion did not affect neurogenesis during cerebral organoid formation, CEP290 deletion increased the number of organoids with choroid plexus. The authors also observed cilia with splayed morphology in the ventricles of cerebral organoids. These data are interesting to understand the function of CEP290 in ciliogenesis and brain development.

We thank the reviewer for their positive assessment of our work.

Major Points:

1. The authors showed that 11 out of 43 CEP290 mutant organoids developed choroid plexus, compared to 1 out of 39 control organoids. It is interesting that CEP290 deletion enhanced choroid plexus generation. The authors may include a graph about choroid plexus generation and TTR- stained images of choroid plexus from control cerebral organoids in Figure 2. However, Lancaster et al. (2013) reported the existence of choroid plexus in their cerebral organoids (25 out of 35), whereas only 1 out of 39 control organoids in this study. This discrepancy should be discussed.

- *We thank the reviewer for this comment, as we were also wondering about the relative lack of choroid plexus in our control organoids. We believe that this could be explained by the individual propensity of individual iPSC lines to form such structures. Indeed, it is well known in the iPSC field that different iPSC lines may behave differently with the same protocol, which is likely to become very apparent with this particular unguided protocol, further underscoring the importance of using isogenic lines that are at least genetically as close to each other as possible. It is well possible that the control HMGU1 clone 5 line has baseline differences in the strength of signaling pathways compared to the iPSC lines used in the original Lancaster paper, which could explain the lack of choroid plexus when applying this protocol to this line. In the isogenic CEP290 mutant lines, these pathways could be slightly affected by the mutation, explaining the increased propensity to form choroid plexus. As discussed below, the scRNA sequencing data is unfortunately not strong enough to prove (or disprove) this hypothesis due to technical limitations (i.e. the quasi absence of choroid plexus cells in the controls to make a fair comparison - see discussion below). We have added a short discussion in the manuscript (but given the space limitations, we had to remain superficial on this). Concerning adding an image of TTR-stained choroid plexus in controls, we were not able to generate one (as explained above for reviewer: the quasi absence of choroid plexus in controls leads to the fact that all slides we stained for TTR did not contain such structures. The scoring was performed on a much larger set of slides/sections with various stainings and based on morphological appearance (which was then confirmed in mutants with TTR staining)).*

2. Did the choroid plexus contain cilia with abnormal morphology in the CEP290 mutant organoids?

- *We are reluctant to make claims about this point - in the immunofluorescence images we have generated, there might be some hint for slightly enlarged ciliary tips, but this is not clear. Since choroid plexus regions are however present in only a subset of the slides (and since we had not stained most slides with TTR previously and did not see such structures in the slides used for expansion microscopy), we cannot perform a systematic analysis and would be reluctant to make any statements about the*

morphology of cilia in the choroid plexus in mutants (especially since we have virtually no control samples to compare with). Below is one example of cilia in mutants in TTR positive regions.

3. The authors mentioned that Wnt signaling is not changed in the choroid plexus cluster based on single cell RNA-seq data. Can the authors show the Wnt signaling data of the choroid plexus cluster in the supplementary figure? Were other signaling pathways—such as Shh, BMP, or mTOR—differentially expressed in this cluster? I understand that single cell RNA-seq data is not easy to interpret when the number of cells in the cluster is too low. If the authors think it is appropriate not to show the data to avoid confusing readers, that is understandable.

- *Like the reviewer suggests, we were hoping to identify pathways that are differentially expressed in the mutant cells of cluster 7 (choroid plexus) compared to controls.*

Unfortunately, as the reviewer points out, we found only a very small number of cells in cluster 7 in controls (n=34 cells), such that we feel that the statistical analyses are limited and that the signals that we could see are not strong enough to draw confident conclusions.

- *Below we show different analyses we have performed:*

- o *Panel (A) (below): for a global unbiased assessment of various pathways we have used the PROGENy tool (<https://saezlab.github.io/progeny/articles/ProgenySingleCell.html>, Schubert et al, 2018, Nature communications, <https://doi.org/10.1038/s41467-017-02391-6>), which takes into account known targets of signaling pathways and integrates the knowledge of them being positive or negative pathway regulators with them being up- or downregulated to provide a global and unbiased evaluation of the state of a pathway. The heatmap shows which pathways are deemed more active in each cluster (in red) and which are less active (blue). With these analyses, it appears that Wnt signaling may be more active in cluster 7 than in other clusters. However, several other pathways also appear to be more active in this cluster.*
- o *Panel (B): To follow-up on this, we therefore next determined which Wnt pathway components (based on KEGG database) are significantly more highly expressed in cluster 7 compared to the other clusters (B, adjusted p-value < 0.001 and log2FC > 0.5). As you can see from this plot, while there few are of these genes more highly expressed in cluster 7, they are not necessarily specific for WntT signaling (for example SMAD3).*
- o *Panel (C): We next compared the expression levels of these genes from (B) in mutant versus control cells from cluster 7 - please note the limitation that there are only very few control cells in cluster 7. In this analysis only a few genes stood out as being upregulated in mutant versus control cells, but the interpretation of this result is limited.*
- o *Panel (D): to further make sure that we were not missing evidence for activated WNT signaling, we next chose known WNT pathway components and compared their expression between mutant and control cells in cluster 7. No striking increase in expression of WNT pathway components was noted.*
- o *Panel (E): we further performed the same comparison, this time analyzing WNT target genes taken from the WNT homepage from the Nusse lab at Stanford university (https://wnt.stanford.edu/target_genes).*

- o *Panel (F): Finally, since HH signaling is not part of the PROGENy analysis, we also analyzed components of this pathway in a targeted manner with similar analyses and saw no evidence for altered HH signaling in mutant versus control, or increased WNT signaling in cluster 7 compared to other clusters*
- *Taken together, we do not feel that these data are strong enough to be included as supplementary figures, as we want to avoid overinterpretation of the data (since there are some hints towards increased WNT signaling in cluster 7 compared to other clusters, and no evidence for increased Wnt signaling in mutant vs control). We fear that showing plots may be misleading and have therefore only added xls spreadsheets as supplementary data with all results from these analyses. We also briefly mention discuss this in the main text.*
- *Please also see our response to reviewers 1 and 3 about further analyses performed on the scRNA sequencing data.*

A- PROGENy pathway analysis per cluster: WNT (and other pathway(s) are more active in choroid plexus cluster (7)

B - WNT pathway genes (based on KEGG) that are more highly expressed in cluster 7

C - WNT components more highly expressed in cluster 7: comparison Mutant vs Control in cluster 7

D - canonical WNT components comparison Mutant vs Control in cluster 7

E - WNT pathway targets (gene selection from Nusse lab website): comparison Mutant vs Control in cluster 7

F - HH pathway analysis

Mutant vs control cluster 7

Mutant cells across clusters

4. Could the authors perform a western blot of CEP290 on CEP290 KO iPSC cells?

- *The CEP290 WB on both mutant iPSC lines is published in the resource paper describing the generation of the lines (Figueiro-Silva et al, <https://doi.10.1016/j.scr.2025.103781>). We have now added to this manuscript the CEP290 WB of both lines in organoids, showing no difference compared to iPSCs - almost complete absence of CEP290 protein in both mutants.*

5. In Figure S6, the authors showed GFAP-positive cells. However, the image quality is too low to determine whether these cells are astrocytes or background. I want to see high-mag images. In addition, are these GFAP-positive cells also positive for S100B?

- *We have added a new supplementary figure (S6) with high resolution S100B and GAD65 images*

Minor Comments

1. Please include the catalogue numbers of reagents in the Methods section.
- *Have been added*
2. Western blot protocol is not written in the Methods section.
- *Has been added*
3. Nanog is listed in the Methods section and the legend of supplementary Figure 7, but no Nanog data were shown.
- *Thank you for noticing this; we have removed this mention (we had used nanog for the QC of the iPSC lines, published now in Figueiro-Silva et al, <https://doi.org/10.1016/j.scr.2025.103781>).*
4. In the ultrastructure expansion microscope experiments, when and how was immunostaining performed? Was the protocol, such as antibody concentration and incubation condition, similar to those used for "Histology and IF of cerebral organoids"?
- *The staining was performed after expansion. We have added detailed information in the methods section*
5. In Figure 1A, asterisks are hiding the ventricles in the organoids. Please use arrows instead of asterisks.
- *We have made this change.*
6. In Figures 1C and 1D, it's difficult to understand which regions were stained. Please include high-mag images of the ventricular zones and the cortical plate zones, respectively.
- *We have added to figure 1 high resolution confocal images as requested.*
7. In Figure 2A, please specify whether the data were from day 79 or day 80.
- *Thank you for catching this typo; the scRNA sequencing was performed at day 79 (while most immunofluorescence stainings were performed at day 80).*
8. In Figure 2C, I think the mutant bar is narrower than the WT bar.
- *Thank you for noticing this, we have corrected it in the figure.*
9. In the legend for Supplementary Figure 4, TBR1 is labeled as a subventricular zone (SVZ) marker.
- *We have modified this.*
10. In Supplementary Figure 4, please provide high-mag images of TBR1 and CTIP2 staining to assess the co-localization.
- *This has been added to the new main figure 1.*
11. In Supplementary Figure 5A, CEP164 and ARL13B are difficult to distinguish due to the red and magenta colors. Please use a different color combination.
- *We have changed the colors as requested.*
12. In Supplementary Figures 7D and 7E, please provide more explanation. What do "C+" and "C-" mean? Which band is CEP290 in 7E? What is A45468? What does clone C3 refer to?
- *Please excuse the confusion; we have clarified the labels.*
13. In Supplementary Figure 8F, there is a stray straight line on the left side of the "normal" bar.
- *Thank you for noticing this. We have removed this bar.*
14. In the Ethics statement, the protocol number for human research approval was not provided. If available, please include it.
- *Done*

**Reviewer 3: SUMMARY OF THE ADVANCE MADE IN THIS PAPER AND ITS POTENTIAL SIGNIFICANCE TO THE FIELD
SUGGESTIONS TO AUTHORS**

Eschment et al, in this work, have generated human brain organoids and studied a previously unexplored area of cilia abnormalities in brain organoids due to mutations in Cep290, a ciliary gene implicated in TZ formation for proper ciliary function. In this aspect, the work has shown that the cilia are bulged at the distal tip, possibly accumulating IFT proteins, which could cause defects in ciliary functionality. It is a straightforward and descriptive work that will be helpful to the cilia community. The cilia data presented are impressive and robust. The following points should be carefully addressed before publication. Addressing these comments will enhance the manuscript and make it more attractive to the community.

We thank the reviewer for their positive assessment of our work.

In the abstract, it reads that ciliopathies are often connected to brain disorders, a sentence that is not entirely true. On the other hand, there are cilia mutations leading to neurodevelopmental disorders (although centrioles are structurally and numerically normal). As the paper addresses brain organoids that indicate neurodevelopmental disorders, it is worthwhile to discuss "non-syndromic" ciliopathies, which encompass brain developmental disorders that do not affect other organs. As more evidence emerges, it is critical to emphasize the aspect of neurodevelopmental disorders due to ciliary dysfunctions as non-syndromic ciliopathies.

- *The reviewer is perfectly right to point out the association between neurodevelopmental disorders and ciliary dysfunction. We believe that the wording used in the abstract (that ciliopathies often affect the central nervous system) is appropriate for a short abstract (based on CNS phenotypes in JBTS, MKS, ACS, etc). Moreover this is a short report limited to 3000 words, where results and discussion are merged, such that the space is very limited to discuss the role of cilia in the CNS more in depth and we unfortunately cannot add more discussion on this topic unless we are allowed a larger word count.*

As the work majorly deals with defects in neural development in organoids differentiated in an undirected manner, call them brain organoids. In which aspect of the organoid protocol has been changed? Clarify it. The scheme shown is confusing, as if SOX2, Nestin, and other NPC markers are provided to the culture at a specified period. I suggest removing the scheme and providing any details in text.

- *We have modified the schematic to clarify, with a detailed supplementary figure legend.*

The authors have chosen undirected differentiation with the reasoning that directed differentiation may cause defects affecting primary cilia-mediated signaling. While it is intriguing, is there any citation that proves these? Cite them.

- *What we meant is that many directed differentiation protocols often use molecules that affect hedgehog signaling (such as SAG) or WNT (such as CHIR) signaling (reviewed for example in Mayhew and Singhania, STAR protocols 2023, <https://doi.org/10.1016/j.xpro.2022.101860>) and since both pathways have been clearly shown to be affected by ciliary dysfunction in many settings, including in CEP290 mutant conditions (<https://doi.org/10.1038/s41598-018-35614-x>, <https://doi.org/10.1073/pnas.1322373111>, <https://doi.org/10.1371/journal.pone.0325921>), we preferred to choose a protocol with minimal intervention using small molecules. We have adapted the wording in the manuscript to clarify.*

Figure 1A: Day 8 organoid cultures are referred to as EBs. Clarify this. EBs do not show neuroepithelium, and there is no characterization of the EBs that has been presented (ectoderm, mesoderm, and endoderm). Clarify whether they are day 8 organoids or EBs. It appears they are day 8 cultures. EBs appear well before differentiation is induced.

- *The denominations we had used were based on the original paper from Lancaster et al. We agree however that technically an EB should contain all the three germ layers and have therefore changed the wording as requested.*

Additionally, most panels display only one object, which raises concerns about the statistical power and robustness of the data. Display a group of organoids or EBs in one panel.

- *We have added a supplementary figure S8 showing groups of organoids, modified the display of the quantification plot and added the raw data of organoid sizes and analyses as supplementary table S1.*

FACS analysis of cell cycle stages is an interesting approach. I am curious about the analysis and which surface marker was used to sort fluorescently? Unless it is a flow cytometry.

- *Apologies for this mistake, this is of course flow cytometry.*

Figure 1, showing the organoid morphologies, is unfortunately of low resolution and does not reveal cytoarchitectures. This is often the case with methods adapted from reagents related to Stem Cell technology. In this manner, I fail to see apicobasal polarity between NPCs and cortical neurons. Show a particular region (cropped with high magnification) to solve this uncertainty. The same is true for GAD65 and GFAP. Show it if not; remove these panes, as GFAP is not specific. At

the same time, TBR1 and CTIP2 stainings are very good. I would recommend that they adapt a zoomed-in image of a particular region and incorporate it into the central figure.

- *We have added high resolution images as requested, which clearly show the layering of the units with progenitor cells around the lumen and neurons away from it.*

The Western blot for validating the clones in Figure S7 lacks the bands in one blot, the loading control, and appears to be a compilation of a few blots. This is crucial to address to prevent severe problems that may arise in the future.

- *We have replaced this image with a new complete blot containing the loading control.*

All iPSC lines (used to generate brain organoids, including HMGU1 lines) should be validated to demonstrate that they have Cep290 deficiency.

- *The isogenic HMGU1 iPSC lines (clone 5 and CRISPR'd clones) have now been published (Figueiro-Silva et al, <https://doi.10.1016/j.scr.2025.103781>), containing complete characterization as per ISSCR guidelines. In the current manuscript we have now added the western blot at the organoid stage (unchanged compared to iPSC stage).*

There is no such thing as ventricular cilia. Ventricular Zones (again, this must be specified in the figures as VZ) is a place where neurogenesis begins. Apically, it forms a lumen called the ventricular lumen. Cells emanate primary cilia into this lumen. Ventricular lumen is far away from the cortical plate and cannot be named as the cortical lumen. Correct these terminologies.

- *We have adapted the nomenclature as requested.*

The paper overall lacks validation for iPSCs (in terms of showing pluripotency markers) and quantification.

- *Complete validation of the HMGU1 CRISPR'd iPSC lines is found in Figueiro-Silva et al (<https://doi.10.1016/j.scr.2025.103781>) and for the patient line in supplementary figure S12.*

Provide quantifications for the VZ thickness and the primitive cortical plates.

- *We have added this information. There is a minimally statistically significant decrease in primitive cortical plate thickness in the Mutants, which could be consistent with the trend for fewer more differentiated neurons in the scRNA sequencing data. Since these differences are however rather subtle and variable across organoids from different batches, we would be reluctant to make strong claims on this. We have added a sentence in the manuscript to mention these findings, remaining conservative in our interpretations.*

scRNA sequencing is a low-resolution experiment and, of course, will not reflect defects or misregulated components related to signaling. However, since the data is there, the author takes all signalling components and tests their distribution across various cell clusters, which may reveal localized signals in the Choroid plexus cluster.

- *Thank you for this suggestion, which we have implemented (please also see the detailed response to reviewer 2 with the graphs showing the analysis of different signaling pathways using the PROGENy tool). Unfortunately, as explained in our response to reviewer 2, we did not find strong enough evidence for an alteration of any of the pathways, which, as the reviewer mentions, is likely explained by the low resolution of the experiment.*
- *Below we display additional violin plots comparing different pathways across clusters. The components of each pathway were selected from the KEGG database (except for Wnt, where the left plots are from a targeted analysis and only the right plots are from KEGG), such that the same gene can appear in more than one pathway:*

Wnt (targeted analysis)

mTOR

Wnt (KEGG - significantly higher expressed in cluster 7)

Hippo

Additionally, it will be crucial to display the cilia component distribution across various clusters and present them as quantitative dot plots. In this respect, the authors can consider

Cep290 interacting components and genuine ciliary molecules.

- We have analyzed the expression of various ciliopathy genes from different ciliary modules/subcompartments as well as known CEP290 interacting proteins and overall see no significant alteration when comparing mutant and control cells (across all cells or only in cells from cluster 7) or when comparing mutant cells in cluster 7 to all other clusters. Please see the violin plots below. However, as also discussed above and in the response for the other reviewers, the generally very low expression level of ciliary genes and the statistical limitations (for example small number of control cells in cluster 7) do not allow to draw very strong conclusions and we would be reluctant to add these plots in the manuscript to avoid overinterpretation of the data. We have added some mention of these analyses as well as the xls spreadsheets with the results as supplementary data files (Supplementary Table S5), so interested readers can judge by themselves. The original raw data has also been uploaded on the Gene Expression Omnibus (GEO) for further analyses for other groups.

Selected ciliary genes across clusters

Second decision letter

MS ID#: jcs.264092R1

MS Title: CEP290-deficiency disrupts ciliary axonemal architecture in human iPSC-derived brain organoids

Authors: Melanie Eschment; Olivier Mercey; Ellen Aarts; Ludovico Perego; Joana Figueiro-Silva; Affef Abidi; Michelle Mennel; Melanie Generali; Anita Rauch; Paul Guichard; Virginie Hamel; Ruxandra Bachmann-Gagescu

Article Type: Short Report

Dear Ruxandra and team,

Happy Friday- I am thrilled to inform you that your revised manuscript has been accepted for publication in our Special Issue on Cilia and Flagella for the Journal of Cell Science, subject to these minor edits listed below and pending standard publication integrity checks. It was accepted on 19 Sep 2025. Where referee reports on this version are available, they are appended below.

The minor edits we would like to see done would be:

The avoid claims 'of being observed in other cell types previously'- Claims of novelty are generally best avoided as they are difficult to prove in an ever expanding field of literature and trigger
 © 2025. Published by The Company of Biologists under the terms of the Creative Commons Attribution License (<https://creativecommons.org/licenses/by/4.0/>).

debates like the one with Reviewer 1. I would suggest that the authors change the abstract sentence 'Such ciliary abnormalities have not been associated with CEP290-dysfunction yet and may represent a tissue-specific consequence revealed by studying a human neuronal organoid model' to 'Such ciliary abnormalities may represent a tissue-specific consequence revealed by studying a human neuronal organoid model'- without losing any impact. Similarly from the Summary Statement change 'Ciliopathy disease modeling using hiPSC-derived brain organoids reveals previously unrecognized defects in ciliary microtubule architecture secondary to deficiency of the transition zone protein CEP290.' to 'Ciliopathy disease modeling using hiPSC-derived brain organoids reveals defects in ciliary microtubule architecture secondary to deficiency of the transition zone protein CEP290.' There is a lovely and complementary piece in this issue for CEP290 in mouse and human photoreceptors from Moyes and Wensel focusing on microtubules, membranes and TZ using an array of advance imaging which I see as converging on similar messaging to your story. If possible, an IF-stained panel is included to document this absence of choroid plexus in most control WT cultures. Consistency in nomenclature for the terms "WT" and "HGMU Clone 5" throughout the text.